# VIP5: Towards Multimodal Foundation Models for Recommendation

**Shijie Geng, Juntao Tan, Shuchang Liu, Zuohui Fu, Yongfeng Zhang**
Department of Computer Science, Rutgers University, NJ 08854, US
{sg1309, juntao.tan, shuchang.syt.liu, zuohui.fu, yongfeng.zhang}@rutgers.edu

## Abstract

Computer Vision (CV), Natural Language Processing (NLP), and Recommender Systems (RecSys) are three prominent AI applications that have traditionally developed independently, resulting in disparate modeling and engineering methodologies. This has impeded the ability for these fields to directly benefit from each other's advancements. With the recent development of foundation models, large language models have emerged as a potential general-purpose interface for unifying different modalities and problem formulations. In light of this, we propose the development of a multimodal foundation model (MFM) considering visual, textual, and personalization modalities under the P5 recommendation paradigm, thus named VIP5 (Visual P5), to unify various modalities and recommendation tasks. This will enable the processing of multiple modalities in a shared architecture for improved recommendations. To achieve this, we introduce multimodal personalized prompts to accommodate multiple modalities under a shared format. Additionally, we propose a parameter-efficient training method for foundation models, which involves freezing the P5 backbone and fine-tuning lightweight adapters, resulting in improved recommendation performance and increased efficiency in terms of training time and memory usage. Code and data of VIP5 are available at `https://github.com/jeykigung/VIP5`.

## 1 Introduction

With rapid growth, recommender systems have gradually become an indispensable element in people's daily lives. With more time spent on the Web, people reveal their interests through richer modalities than before. In response to the trend, current recommendation systems (Meng et al., 2020; Hou et al., 2019; Zhang et al., 2021a, 2017) consider more diverse contents when making recommendation decisions to users.

Historically, the technical developments for processing different types of information (such as personalization, visual, and textual) are mostly spread across different research communities. Fortunately, recent advances in Foundation Models (FMs) such as Large Language Models (LLMs) unfold a promising route for building general-purpose models and unifying diverse modalities, so that one single architecture can handle visual, textual and personalized information at the same time, enabling the possible approach towards Artificial General Intelligence (AGI) (Ge et al., 2023) and Artificial General Recommender (AGR) (Lin and Zhang, 2023). As a pioneering work, GPT-3 (Brown et al., 2020) can perform in-context learning, enabling it to solve brand-new problems given few-shot demonstration examples as prompts. Similarly, CLIP (Radford et al., 2021; Geng et al., 2022d) maintains superior zero-shot generalization ability when shifting to an out-of-distribution visual domain if provided with appropriate prompt. With more and more emergent abilities (Wei et al., 2022b) revealed in foundation models, they become not only a popular backbone to finetune downstream tasks (Alayrac et al., 2022; Sanh et al., 2022; Wei et al., 2022a) but also an effective training scheme for unifying multiple modalities in a shared interface (Wang et al., 2022; Chen et al., 2022; Cho et al., 2021; Jiang et al., 2022). Following the trend in language and vision domains, P5 (Geng et al., 2022c) and M6-Rec (Cui et al., 2022) put forward the concept of personalized foundation models for recommendation and propose to pre-train LLMs on instructional prompts to accommodate various recommendation tasks under a shared model and training objective.

While there are large models for language (Raffel et al., 2020; Brown et al., 2020), vision (Yu et al., 2022; Radford et al., 2021), graphs (Ye et al., 2023; Geng et al., 2022b) and recommendation (Geng et al., 2022c; Cui et al., 2022) domains

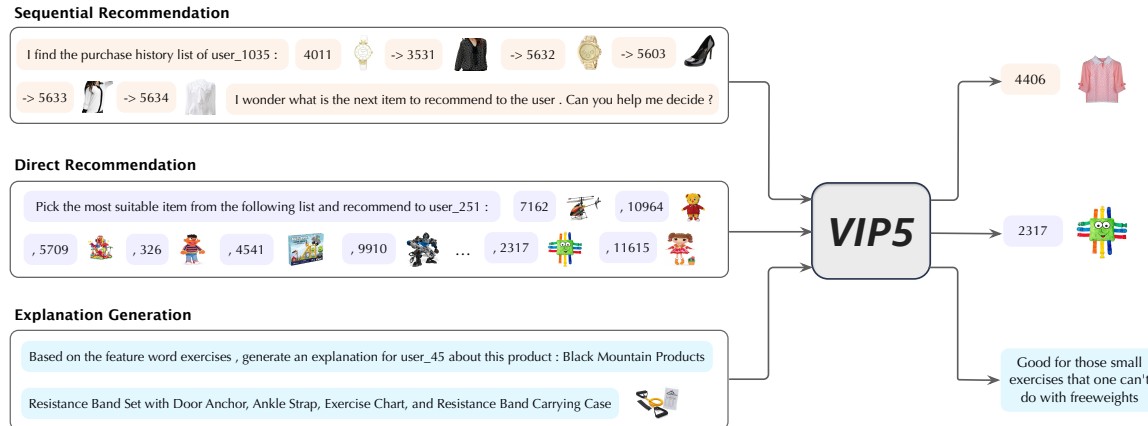

Figure 1: An example task scope of VIP5 covering three popular recommendation tasks. Based on multimodal personalized prompts (*left*) that interleave language and visual tokens, VIP5 is able to transfer all task and all modalities into a unified sequence format, and generates target outputs (*right*) according to certain task descriptions. VIP5 treats large language models as a fixed general-purpose interface and finetunes extra visual and language processing layers to achieve the ability for handling various recommendation tasks.

separately, in this work, we take one step further and aim to unify the above foundation models to jointly process multi-modality information sources for personalization and recommendation. To this end, we follow the "Pretrain, Personalized Prompt and Predict Paradigm (P5)" for recommendation (Geng et al., 2022c; Xu et al., 2023) and propose a Multimodal Foundation Model (MFM) named VIP5 (Visual P5), which provides the following advantages for recommender systems: 1) VIP5 provides multimodal personalized prompts for supporting all modalities' connections to the recommendation foundation model. Specifically, to construct multimodal personalized prompts, VIP5 employs a mapping network to transfer features from other modalities into the corresponding tokens. By this step, multimodal features are projected to the same manifold space of the backbone foundation model. 2) The VIP5 framework provides the ability of *Parameter-efficient tuning* rather than *Pretraining* in existing recommendation foundation models such as P5 (Geng et al., 2022c). Different from the pre-training step of P5 that updates all the parameters in the backbone foundation model – which is impractical when the size of foundation model grows explosively – VIP5 only finetunes a small proportion of extra lightweight adapter modules during training while maintaining the large language model backbone fixed. 3) With the ability of multi-modality learning and parameter-efficient tuning, VIP5 further improves the performance of recommendation foundation models with both less training time and less memory usage, making it easier to train and deploy foundation models for recommendation. Overall, our key contributions

are outlined as follows:

- We propose VIP5 framework to unify CV, NLP, and RecSys foundation models and facilitate recommendation with multimodal information.
- We introduce multimodal personalized prompts to adapt multi-modality information into a shared tokenized space with textual, visual and personalization inputs.
- We develop adapter-based parameter-efficient tuning for VIP5 to achieve a better recommendation performance and training efficiency.
- Based on the experimental results, VIP5 beats strong baselines on three task groups while saving substantial training time and memory usage.

## 2 Related Work

**Prompt Learning.** Prompt learning (Liu et al., 2021) gradually emerges as a popular paradigm to control the behavior of large language models since it can effectively adapt a pretrained model to downstream tasks in either zero-shot or few-shot style. The success of GPT series (Radford et al., 2019; Brown et al., 2020) attracts the first wave of interests on the topic. The in-context learning capability of GPT-3 (Brown et al., 2020) inspires many efforts on automatic prompt search or generation (Gao et al., 2021; Jiang et al., 2020; Shin et al., 2020; Zhang et al., 2022) to achieve higher-quality discrete prompts. However, it is naturally hard to optimize these approaches in a discrete space. To solve this issue, soft prompt based approaches such as Prefix-Tuning (Li and Liang, 2021), Prompt-Tuning (Lester et al., 2021), CoOp (Zhou et al., 2022), and Visual-Prompt Tuning (Jia et al., 2022)

are proposed to leverage additional trainable continuous embeddings as prefix to conduct finetuning on downstream tasks. While achieving better scalability and generalization ability, the learned soft prompts are more difficult to interpret than discrete prompts. To accommodate all above merits, instruction prompts that directly describe different tasks via natural language instructions are adopted by a lot of methods (Weller et al., 2020; Wei et al., 2022a; Sanh et al., 2022; Aribandi et al., 2022; Mishra et al., 2022), highlighting significant improvements on unseen tasks.

**Large Recommendation Models.** Motivated by the success of Large Language Models (LLMs), the RecSys community started to pay more attention to recommendation model's generalization ability and transferability (Li et al., 2023b). For example, inspired by the prompt learning paradigm, PETER (Li et al., 2021) and PEPLER (Li et al., 2022) proposes to learn personalized continuous prompts to represent user and item IDs and generates natural language explanations to justify recommendations. In contrast, M6-Rec (Cui et al., 2022) converts all user behavior information to plain text sequences and feeds them into a Transformer encoder, and then designs a task-specific training loss for downstream tasks and finetuning. Apart from previous efforts, P5 (Geng et al., 2022c) and OpenP5 (Xu et al., 2023) leverage not only instruction-based finetuning on LLMs to represent personalized fields for users and items but also unifies various tasks via natural language instructions. Hence, P5 is able to unify various recommendation tasks into a shared encoder-decoder architecture and a joint training objective. P5-ID (Hua et al., 2023b) further explores different item ID creation methods for LLM-based recommendation models, such as sequential indexing, collaborative indexing, semantic indexing, and hybrid indexing.

**Multimodal Recommendation.** Current approaches to multimodal recommendation can be divided into three categories. The most common usage is to leverage multimodal content as side information to assist recommendation decisions. For example, VBPR (He and McAuley, 2016) proposes using visual features to supplement user feedback and improve matching-based recommendations. PiNet (Meng et al., 2020) proposes to cover more personalized visual preferences about users. It simultaneously learns heterogeneous visual features with semantic and collaborative information and then fuses different visual information through a dual-gating module. JRL (Zhang et al., 2017) proposes Joint Representation Learning over multiple modalities for improved recommendation. Another stream of approaches focus on providing recommendations along with correlated visual explanations. These methods usually work in domains where visual information is important to user behavior patterns, such as fashion (Hou et al., 2019; Verma et al., 2020; Chen et al., 2019), travel (Geng et al., 2022a), and food (Meng et al., 2020). Furthermore, several recent approaches have been proposed to discover the rich intra-item and inter-item semantic structures from multimodal contents to facilitate better item representations and thus enhance recommendation performances (Zhang et al., 2021b,a; Deldjoo et al., 2022).

# 3 VIP5 Paradigm with Multimodal Personalized Prompts

We introduce the proposed VIP5 paradigm in this section. In Section 3.1, we incorporate multimodal signals into personalized prompts. In Section 3.2, we elaborate how to conduct parameter-efficient tuning with adapters based on multimodal personalized prompts.

## 3.1 Multimodal Personalized Prompts

A personalized prompt includes personalized fields for users and items (Geng et al., 2022c; Li et al., 2022, 2023a), with formats ranging from ID numbers to detailed descriptions. In our work, we develop foundation models as a general-purpose interface to connect available modalities that could be helpful for eliciting user preferences. To facilitate this end, we propose "multimodal personalized prompts". Technically, we consider textual, visual, and personalization information as three example modalities in our multimodal personalized prompts (Figure 2).

Given an item image $\mathbf{I} \in \mathbb{R}^{H \times W \times 3}$, where $H$ and $W$ are the image height and width, we first adopt a visual encoder such as CLIP image branch (Radford et al., 2021) to extract its feature $x \in \mathbb{R}^{d_v}$, where $d_v$ represents the visual feature dimension. To connect the image feature to other text-based tokens in a personalized prompt, as illustrated in Figure 2(c), we design a mapping network $f$ with two linear layers to transfer the original image feature to $k$ image tokens: $p_1, \ldots, p_k = f(x)$. Then we append the image tokens to their corre-

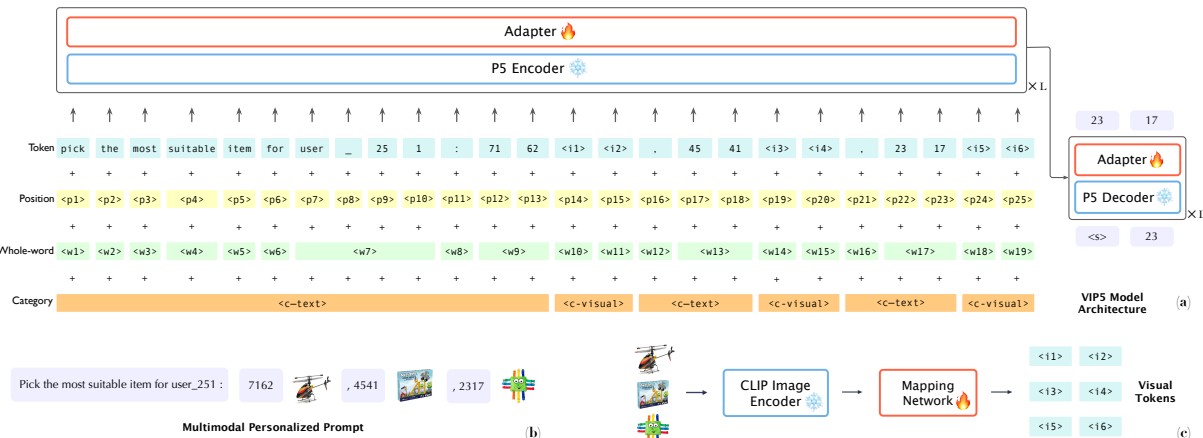

Figure 2: An illustration of the VIP5 framework. VIP5 is built on an encoder–decoder Transformer model that takes in textual inputs as well as image inputs to produce responses or make recommendation decisions. In the figure, a fire symbol represents training with parameter update while a snow flake symbol stands for the frozen parameters.

sponding item tokens to construct a multimodal personalized field $\mathcal{M}$:

$$\mathcal{M} : \underbrace{w_1 \cdots w_m}_{\text{item tokens}}, \underbrace{p_1, \ldots, p_k}_{\text{image tokens}}. \qquad (1)$$

We create a collection of 29 multimodal personalized prompts covering three important task families – *sequential recommendation*, *direct recommendation*, and *explanation*. The full list of prompts is provided in Figure 8, 9, and 10. Based on the collection of multimodal personalized prompts, we use the multimodal personalized field $\mathcal{M}$ as in Eq.(1) to substitute the item field in the prompt. It is worth noting that the prompts for sequential and direct recommendation usually contain more than one multimodal personalized fields.

### 3.2 Parameter-efficient Tuning with Adapters

Our VIP5 framework is shown in Figure 2(a). For tokenized multimodal sequence $\mathbf{S}$, we first apply position encoding $\mathcal{P}$ and whole-word embedding $\mathcal{W}$ on $\mathbf{S}$ to help the model better recognize the absolute positions of input tokens and important user/item fields (e.g., "user_251" is split into 4 separate tokens ["user", "_", "25", "1"], but they share the same whole-word embedding "$\langle w7 \rangle$"). Besides, we adopt an additional category embedding $\mathcal{C}$ to identify whether a token is textual or visual. Afterwards, we feed the resulting sequence into the $L$-layered text encoder $\mathcal{E}$ and decoder $\mathcal{D}$ modules.

Besides multimodal personalized prompts, we propose parameter-efficient tuning using adapters for computation- and memory-efficient training. By inserting adapters into the foundation model backbone, freezing its parameters, and updating lightweight adapter modules, this strategy largely

reduces trainable parameters, decreasing training time and memory usage. Tuning few additional parameters addresses efficiency concerns when incorporating visual tokens into text-based personalized prompts. More importantly, fine-tuning the entire backbone may cause over-fitting for easier tasks, whereas parameter-efficient tuning can leverage both training efficiency and the power of large foundation models.

Formally, if we denote the input sequence for the $i$-th layer of text encoder as $\mathbf{S}_i = [s_1, \cdots, s_n]$, in traditional Transformer, $\mathbf{S}_i$ will go through one self-attention block and a feed-forward network. While in VIP5, we insert adapters (Houlsby et al., 2019; Sung et al., 2022) in both the self-attention block and the feed-forward network, the exact position is after each module and before the LayerNorm (Ba et al., 2016). The whole process can be written as:

$$\mathbf{S}_{i+1} = A_2 \left( \text{FFN} \left( A_1 \left( \text{Attention} \left( \mathbf{S}_i \mathbf{W_Q}, \mathbf{S}_i \mathbf{W_K}, \mathbf{S}_i \mathbf{W_V} \right) \right) \right) \right),$$
$$(2)$$

where $\mathbf{W_Q}, \mathbf{W_K}, \mathbf{W_V} \in \mathbb{R}^{d \times d_h}$ are weight matrices for projecting query, key, and value, respectively, $d_h = d/h$ is the dimensionality for each head. The Attention function is defined as:

$$\text{Attention}(\mathbf{Q}, \mathbf{K}, \mathbf{V}) = \text{softmax} \left( \frac{\mathbf{Q}\mathbf{K}^\top}{\sqrt{d_h}} \right) \mathbf{V}. \qquad (3)$$

Besides, FFN is a feed-forward module consisting of two fully-connected layers. $A_1$ and $A_2$ are the feature adapters after the self-attention and feed-forward network. They are both bottleneck fully-connected layers with an activation function in between. We can represent these adapters as:

$$A = f_{\text{up}} \left( \sigma \left( f_{\text{down}}(\mathbf{S}_i) \right) \right) + \mathbf{S}_i, \qquad (4)$$

where $f_{\text{down}}$ and $f_{\text{up}}$ are the down-sampling and up-sampling layers of an adapter, and $\sigma$ is the GELU activation function (Hendrycks and Gimpel, 2016). Similar to text encoder, we also adopt adapters between the cross-attention block and its LayerNorm layer inside text decoder.

VIP5 utilizes the conditional token generation loss for all three recommendation tasks. After encoding the input multimodal personalized prompts into a contextualized latent sequence with $\mathcal{E}$, the text decoder $\mathcal{D}$ autoregressively predict next tokens conditioned on the already generated tokens $\mathbf{y}_{<j}$ and the input text $\mathbf{t}$. In summary, VIP5 adopts the following training objective to perform parameter-efficient tuning with adapters:

$$\mathcal{L}_\theta = -\sum_{j=1}^{|\mathbf{y}|} \log P_\theta \left( \mathbf{y}_j \mid \mathbf{y}_{<j}, \mathbf{t} \right). \quad (5)$$

After training, we perform inference with VIP5 based on given multimodal personalized prompts. For sequential and direct recommendation task groups, we create a list of candidate items for recommendation via beam search. For explanation task group, we simply apply greedy decoding for text generation.

## 4 Experiments

In this section, we provide the performance comparison between the VIP5 framework and representative approaches for different task groups. We conduct extensive experiments and ablation studies to explore the following research questions:

- **RQ1:** Does the proposed parameter-efficient VIP5 framework perform well when compared with baseline methods across the three task groups?
- **RQ2:** When conducting parameter-efficient tuning, which parts should we insert adapters and perform finetuning? In addition, will different adapter reduction rates affect the performance and efficiency of VIP5?
- **RQ3:** Does visual information play an important role in multimodal personalized prompts? What if we change the number of image tokens and the type of visual encoder?

### 4.1 Experimental Setups

**Datasets.** We employ four real-world datasets collected from *Amazon* platform for experiments

| Dataset | Clothing | Sports | Beauty | Toys |
|---|---|---|---|---|
| #Users | 39,387 | 35,598 | 22,363 | 19,412 |
| #Items | 23,033 | 18,357 | 12,101 | 11,924 |
| #Reviews | 278,677 | 296,337 | 198,502 | 167,597 |
| #Photos | 22,299 | 17,943 | 12,023 | 11,895 |
| #Sparsity (%) | 0.0307 | 0.0453 | 0.0734 | 0.0724 |

Table 1: Detailed statistics of the datasets used in our paper.

and ablation studies, namely *Clothing, Shoes & Jewelry*, *Sports & Outdoors*, *Beauty*, and *Toys & Games*. These datasets[1] contain user purchase records, reviews, item descriptions, and images. Table 1 offers detailed dataset statistics.

**Tasks and Metrics.** In this paper, we cover three recommendation task groups: A) sequential recommendation, B) direct recommendation, and C) explanation generation. We follow the same preprocessing steps and train/validation/test splits as in (Geng et al., 2022c). For sequential recommendation, the last and second last items in each user's interaction history are adopted as test and validation ground-truths, with remaining items as training data. For direct recommendation, we use sequential recommendation's train/validation/test splits to generate 100 candidate lists as in (Zhao et al., 2022). For explanation generation, we adopt an 8:1:1 random split and extract rating explanations using the Sentires library (Zhang et al., 2014).

We evaluate sequential and direct recommendation task groups using Hit Ratio (HR@k) and Normalized Discounted Cumulative Gain (NDCG@k), while explanation generation tasks are assessed using text generation metrics like BLEU and ROUGE. In all tables, **bold** numbers indicate the best approach for each metric.

**Implementation Details.** We utilize the pretrained P5-small checkpoint as VIP5's backbone, as it often outperforms P5-base (Geng et al., 2022c). VIP5's encoder and decoder consist of 6 Transformer blocks, a 512-dimension embedding size, and 8 attention heads. To process visual information, we use CLIP's (Radford et al., 2021) image branch as VIP5's visual encoder and pre-extract image features. Similar to P5, we employ the SentencePiece (Sennrich et al., 2016) tokenizer with a 32,100 vocabulary size to generate sub-word input tokens. By default, the mapping network serves as the image tokenizer in VIP5 and the number of image tokens is set to 2, while the adapters have a reduction factor of 8 for the bottleneck dimension.

For each task group, all multimodal personal-

---

[1] http://jmcauley.ucsd.edu/data/amazon/links.html

| Methods | Sports | | | | Beauty | | | |
|---|---|---|---|---|---|---|---|---|
| | HR@5 | NDCG@5 | HR@10 | NDCG@10 | HR@5 | NDCG@5 | HR@10 | NDCG@10 |
| HGN | 0.0189 | 0.0120 | 0.0313 | 0.0159 | 0.0325 | 0.0206 | 0.0512 | 0.0266 |
| SASRec | 0.0233 | 0.0154 | 0.0350 | 0.0192 | 0.0387 | 0.0249 | 0.0605 | 0.0318 |
| S³-Rec | 0.0251 | 0.0161 | 0.0385 | 0.0204 | 0.0387 | 0.0244 | 0.0647 | 0.0327 |
| P5 (A-3) | 0.0272 | 0.0169 | 0.0361 | 0.0198 | 0.0503 | 0.0370 | 0.0659 | 0.0421 |
| VIP5 (A-3) | **0.0412** | **0.0345** | **0.0475** | **0.0365** | **0.0556** | **0.0427** | **0.0677** | **0.0467** |
| P5 (A-9) | 0.0258 | 0.0159 | 0.0346 | 0.0188 | 0.0490 | 0.0358 | 0.0646 | 0.0409 |
| VIP5 (A-9) | 0.0392 | 0.0327 | 0.0456 | 0.0347 | 0.0529 | 0.0413 | 0.0655 | 0.0454 |

| Methods | Clothing | | | | Toys | | | |
|---|---|---|---|---|---|---|---|---|
| | HR@5 | NDCG@5 | HR@10 | NDCG@10 | HR@5 | NDCG@5 | HR@10 | NDCG@10 |
| HGN | 0.0107 | 0.0071 | 0.0175 | 0.0092 | 0.0321 | 0.0221 | 0.0497 | 0.0277 |
| SASRec | 0.0107 | 0.0066 | 0.0194 | 0.0095 | 0.0463 | 0.0306 | 0.0675 | 0.0374 |
| S³-Rec | 0.0076 | 0.0045 | 0.0135 | 0.0063 | 0.0443 | 0.0294 | 0.0700 | 0.0376 |
| P5 (A-3) | 0.0478 | 0.0376 | 0.0554 | 0.0401 | 0.0655 | 0.0570 | 0.0726 | 0.0593 |
| VIP5 (A-3) | **0.0603** | **0.0564** | **0.0632** | **0.0573** | **0.0662** | **0.0577** | **0.0749** | **0.0604** |
| P5 (A-9) | 0.0455 | 0.0359 | 0.0534 | 0.0385 | 0.0631 | 0.0547 | 0.0701 | 0.0569 |
| VIP5 (A-9) | 0.0569 | 0.0531 | 0.0597 | 0.0540 | 0.0641 | 0.0556 | 0.0716 | 0.0580 |

Table 2: Performance comparison on sequential recommendation.

| Methods | Sports | | | | | Beauty | | | | |
|---|---|---|---|---|---|---|---|---|---|---|
| | HR@1 | HR@5 | NDCG@5 | HR@10 | NDCG@10 | HR@1 | HR@5 | NDCG@5 | HR@10 | NDCG@10 |
| BPR-MF | 0.0314 | 0.1404 | 0.0848 | 0.2563 | 0.1220 | 0.0311 | 0.1426 | 0.0857 | **0.2573** | 0.1224 |
| BPR-MLP | 0.0351 | 0.1520 | 0.0927 | 0.2671 | 0.1296 | 0.0317 | 0.1392 | 0.0848 | 0.2542 | 0.1215 |
| VBPR | 0.0262 | 0.1138 | 0.0691 | 0.2060 | 0.0986 | 0.0380 | 0.1472 | 0.0925 | 0.2468 | 0.1245 |
| P5 (B-5) | 0.0574 | 0.1503 | 0.1050 | 0.2207 | 0.1276 | 0.0601 | 0.1611 | 0.1117 | 0.2370 | 0.1360 |
| VIP5 (B-5) | 0.0606 | 0.1743 | 0.1185 | 0.2539 | 0.1441 | 0.0580 | 0.1598 | 0.1099 | 0.2306 | 0.1327 |
| P5 (B-8) | 0.0567 | 0.1514 | 0.1049 | 0.2196 | 0.1269 | 0.0571 | 0.1566 | 0.1078 | 0.2317 | 0.1318 |
| VIP5 (B-8) | **0.0699** | **0.1882** | **0.1304** | **0.2717** | **0.1572** | **0.0615** | **0.1655** | **0.1147** | 0.2407 | **0.1388** |

| Methods | Clothing | | | | | Toys | | | | |
|---|---|---|---|---|---|---|---|---|---|---|
| | HR@1 | HR@5 | NDCG@5 | HR@10 | NDCG@10 | HR@1 | HR@5 | NDCG@5 | HR@10 | NDCG@10 |
| BPR-MF | 0.0296 | 0.1280 | 0.0779 | 0.2319 | 0.1112 | 0.0233 | 0.1066 | 0.0641 | 0.2003 | 0.0940 |
| BPR-MLP | 0.0342 | 0.1384 | 0.0858 | 0.2327 | 0.1161 | 0.0252 | 0.1142 | 0.0688 | 0.2077 | 0.0988 |
| VBPR | 0.0352 | 0.1410 | 0.0877 | **0.2420** | 0.1201 | 0.0337 | 0.1294 | 0.0808 | **0.2199** | 0.1098 |
| P5 (B-5) | 0.0320 | 0.0986 | 0.0652 | 0.1659 | 0.0867 | 0.0418 | 0.1219 | 0.0824 | 0.1942 | 0.1056 |
| VIP5 (B-5) | 0.0481 | 0.1287 | 0.0890 | 0.1992 | 0.1116 | 0.0428 | 0.1225 | 0.0833 | 0.1906 | 0.1051 |
| P5 (B-8) | 0.0355 | 0.1019 | 0.0688 | 0.1722 | 0.0912 | 0.0422 | 0.1286 | 0.0858 | 0.2041 | 0.1099 |
| VIP5 (B-8) | **0.0552** | **0.1544** | **0.1058** | 0.2291 | **0.1297** | **0.0433** | **0.1301** | **0.0875** | 0.2037 | **0.1110** |

Table 3: Performance comparison on direct recommendation.

ized prompts except the last are used to train VIP5. Prompts A-3/A-9, B-5/B-8, and C-3/C-12 are used for evaluation purpose, with A-3, B-5, C-3 testing model performance under seen prompts and A-9, B-8, C-12 under zero-shot unseen prompts. VIP5 is trained for 10 epochs with a batch size of 36 on four NVIDIA A100 GPUs, using a learning rate of $1 \times 10^{-3}$ and AdamW (Loshchilov and Hutter, 2018) optimizer. As multimodal personalized prompts contain more image tokens, we set input token's maximum length to 1024. During inference, beam size $B$ is set to 20 for sequential and direct recommendation tasks that require generating a list of candidate items.

**Comparison Baselines.** To make performance comparisons, we consider a collection of baselines for each task group. For all three task groups, we include P5 (Geng et al., 2022c) as a baseline to compare with existing foundation models for recommendation. P5 pre-trains all tasks with predefined

text-based personalized prompts via autoregressive language modeling loss and performs inference using greedy decoding or beam search to generate outputs. Additionally, we compare with task-specific approaches. For sequential recommendation, baseline methods include **HGN** (Ma et al., 2019), **SASRec** (Kang and McAuley, 2018), and **S³-Rec** (Zhou et al., 2020). For direct recommendation, we compare with **BPR-MF** (Rendle et al., 2009), **BPR-MLP**, and **VBPR** (He and McAuley, 2016). For explanation generation, we inherent the baselines of P5: **Attn2Seq** (Dong et al., 2017), **NRT** (Li et al., 2017), and **PETER** (Li et al., 2021). When providing a hint feature word as input, PETER becomes its variant **PETER+**, which we also use as an explanation generation baseline.

### 4.2 Performance on Task Groups (RQ1)

In this section, we conduct parameter-efficient tuning for VIP5 on multimodal personalized prompts

| Methods | Sports | | | | Beauty | | | |
|---|---|---|---|---|---|---|---|---|
| | BLUE4 | ROUGE1 | ROUGE2 | ROUGEL | BLUE4 | ROUGE1 | ROUGE2 | ROUGEL |
| Attn2Seq | 0.5305 | 12.2800 | 1.2107 | 9.1312 | 0.7889 | 12.6590 | 1.6820 | 9.7481 |
| NRT | 0.4793 | 11.0723 | 1.1304 | 7.6674 | 0.8295 | 12.7815 | 1.8543 | 9.9477 |
| PETER | 0.7112 | 12.8944 | 1.3283 | 9.8635 | 1.1541 | 14.8497 | 2.1413 | 11.4143 |
| P5 *(C-3)* | 0.6212 | 11.8539 | 2.0707 | 9.0189 | 1.0230 | 14.3242 | 2.0761 | 10.9085 |
| VIP5 *(C-3)* | **1.0639** | **14.8628** | **2.1012** | **11.1059** | **1.2850** | **17.7492** | **2.3482** | **12.9170** |
| PETER+ | **2.4627** | 24.1181 | 5.1937 | 18.4105 | **3.2606** | 25.5541 | 5.9668 | 19.7168 |
| P5 *(C-12)* | 1.3144 | 22.9182 | 4.9976 | 17.1976 | 1.6313 | 24.6267 | 4.9623 | 18.6423 |
| VIP5 *(C-12)* | 2.3003 | **24.4887** | **5.5057** | **18.6610** | 2.8390 | **26.0513** | **6.0159** | **20.4387** |

| Methods | Clothing | | | | Toys | | | |
|---|---|---|---|---|---|---|---|---|
| | BLUE4 | ROUGE1 | ROUGE2 | ROUGEL | BLUE4 | ROUGE1 | ROUGE2 | ROUGEL |
| Attn2Seq | 0.6296 | 11.4588 | 1.2558 | 9.0429 | 1.6238 | 13.2245 | 2.9942 | 10.7398 |
| NRT | 0.4599 | 10.1480 | 0.9720 | 8.2434 | 1.9084 | 13.5231 | 3.6708 | 11.1867 |
| PETER | 0.7204 | 12.1836 | 1.3912 | 9.7084 | 1.9861 | 14.2716 | 3.6718 | 11.7010 |
| P5 *(C-3)* | 0.7569 | 12.2833 | 1.8116 | 9.6023 | 1.4522 | 12.6100 | **3.8144** | 10.1450 |
| VIP5 *(C-3)* | **1.1904** | **14.1685** | **2.0308** | **10.8488** | **2.3241** | **15.3006** | 3.6590 | **12.0421** |
| PETER+ | 3.6204 | 28.4342 | 7.7994 | 22.4059 | **4.7919** | 28.3083 | 9.4520 | 22.7017 |
| P5 *(C-12)* | 1.8811 | 27.7922 | 7.3203 | 21.5462 | 2.6216 | 27.8984 | 9.0076 | 21.6136 |
| VIP5 *(C-12)* | 3.2581 | **28.9059** | **8.5168** | **22.8807** | 3.9293 | **28.9225** | **9.5441** | **23.3148** |

Table 4: Performance comparison on explanation generation (numbers are in percentage %).

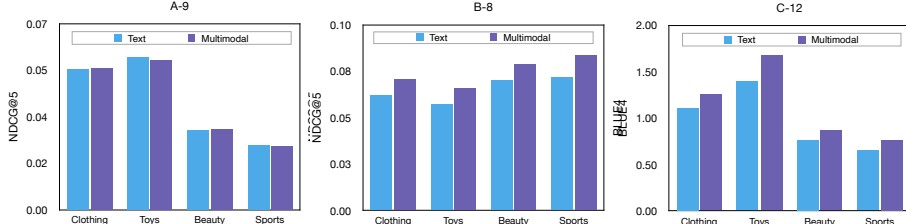

Figure 3: Performance comparison between text-based prompt and multimodal prompt.

from all the three task groups. For each task group, we select one seen and one unseen prompt for evaluation. Performance comparisons with baselines are presented in Table 2, 3, and 4.

**Sequential Recommendation.** As shown in Table 2, we adopt Prompt A-3 and Prompt A-9 to evaluate the performances of different approaches. From the table, we can see that VIP5 is able to achieve better performances than all sequential recommendation baselines on all the four experiment datasets, among which a relatively large gap can be observed on *Sports* and *Clothing* datasets. The results show that our parameter-efficient tuning strategy works effectively on the sequential recommendation task group.

**Direct Recommendation.** For the direction recommendation task group, we evaluate different methods using Prompt B-5 and Prompt B-8 as input multimodal personalized prompts. Table 3 presents the performance comparison, showing VIP5 outperforming all baselines on *Sports*. While VIP5 achieves marginally lower HR@10 on *Toys*, *Beauty*, and *Clothing* datasets, it still surpasses all baselines on other metrics.

**Explanation Generation.** Table 4 illustrates the performance comparison for explanation generation task group. In the table, Prompt C-12 are ap-

plied to evaluate all methods under hint feature word setup, while Prompt C-3 targets at direct explanation generation with only the given user–item pair. The experimental results indicate that VIP5 outperforms other baselines when equipped with the multimodal personalized Prompt C-3. For Prompt C-12, VIP5 achieves superior performances than P5 across all datasets in terms of all metrics and has the highest ROUGE1, ROUGE2, ROUGEL scores of the four experimental datasets.

### 4.3 Parameter-efficient Tuning (RQ2)

In this section, we discuss how to conduct parameter-efficient tuning with adapters to show the impact of different tuning choices.

**How to conduct parameter-efficient tuning.** We first try three fine-tuning approaches: 1) inserting adapters in Transformer's self-attention blocks and only fine-tuning them, 2) fine-tuning adapters in both self-attention and cross-attention blocks, 3) fully fine-tuning all parameters. For this ablation, we conduct experiments on *Toys* with ResNet-101 visual features, a reduction rate of 8, and a single image token in multimodal prompt. Figure 4 demonstrates that fine-tuning adapters in all attention blocks is necessary to achieve better (Prompt C-12) or comparable (Prompt A-9 & B-8) results

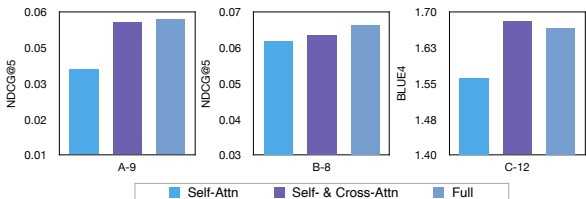

Figure 4: Performance comparison among only activating adapters in self-attention blocks, both self-attention and cross-attention blocks, and full finetuning.

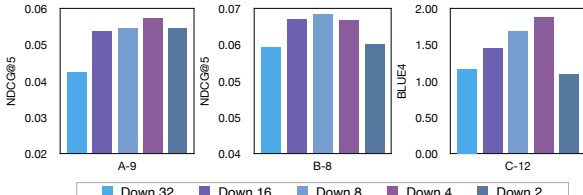

Figure 5: Ablation on the *downsample reduction rate*.

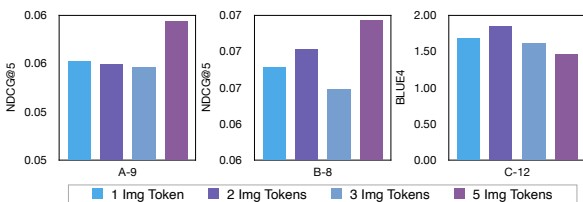

Figure 6: Ablation on the *image token number*.

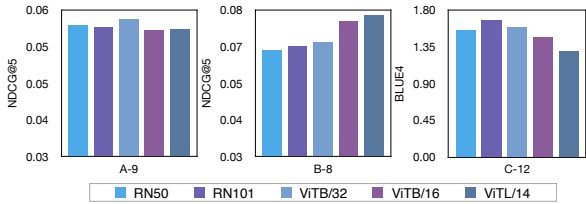

Figure 7: Ablation on the *visual encoder type*.

with full fine-tuning. Moreover, Table 5 shows that the former saves 21.2% time and 18.1% memory usage during training compared to the latter, highlighting VIP5's effectiveness and efficiency.

**On adapter reduction rate.** The reduction rate is an important hyper-parameter for adapters. When decreasing the reduction rate, the hidden dimension of bottleneck layers will increase correspondingly, resulting in a higher percentage of trainable parameters. We select five different values of reduction rates and perform all experiments with ResNet-101 visual features and a single image token in multimodal prompt. In Figure 5, we can see that 4 and 8 are suitable reduction rates for all the 3 task groups.

### 4.4 Ablation on Visual Components (RQ3)

In this section, we aim to explore whether visual information matters for different task groups. We also estimate the influence of the number of image tokens and the visual encoder type.

**Text-based vs. multimodal personalized prompts.** To compare text-based and multimodal personalized prompts, we set the number of image tokens to 0 and 1, respectively, and conduct experiments on all four datasets with a reduction rate of 8 and ResNet-101 visual features. Figure 3 shows that introducing visual signals into personalized prompts improves all datasets for direct recommendation task group (Prompt B-8). This is in line with our expectation that an item's visual appearance significantly influences people's choices. For sequential recommendation, visual information does not bring obvious performance improvements, indicating that the purchase sequence itself is more significant for predicting next items. For explanation generation, visual information positively impacts all datasets, especially for the *Toys* dataset.

**On the number of image tokens.** To examine the influence of the number of image tokens, we select four different numbers (1, 2, 3, 5) and conduct additional experiments on *Toys* with a reduction rate of 8 and ResNet-101 visual features. According to

Figure 6, enabling 5 image tokens in multimodal personalized prompt achieves the best performance on Prompt A-9 and Prompt B-8, while 2 image tokens perform the best for Prompt C-12. However, longer visual prompt results in more training time (e.g., 5 image tokens take 60.8% more time than 2 image tokens). Therefore, we choose 2 image tokens as default setting considering the trade-off.

**On visual encoder type.** The visual encoder type is another factor influencing multimodal personalized prompt representation. We explore various CLIP visual branch architectures: ResNet50, ResNet101, ViT-B/32, ViT-B/16, ViT-L/14 (in an ascending order of visual encoder ability according to CLIP (Radford et al., 2021)). All experiments are performed on *Toys* with a reduction rate of 8 and a single image token. The results are reported in Figure 7. Similar to our previous conclusions, visual information matters most for direct recommendation, with continuous performance gains when using better visual encoders. However, for sequential recommendation and explanation generation, better visual encoders do not always improve performance. This is most likely because the purchase sequence is more crucial than visual information for predicting the next item in sequential recommendation, leading to similar performances under different visual encoders. For explanation genera-

| Methods/Metrics | Time/Epoch | Trainable Param. | Memory Usage |
|---|---|---|---|
| Self-Attn | 10.55 | 2.97 | 27.4 |
| Self- & Cross-Attn | 11.10 | 3.58 | 29.0 |
| Full (P5) | 14.08 | 100 | 35.6 |

Table 5: Comparison of different training strategies in terms of trainable parameter (%), training time (min), and memory usage (GB) on the *Toys* dataset.

tion, hint words significantly influence generated sentences, and the compatibility between hint word and visual embedding varies across different visual encoders. However, VIP5 is still better than the best baseline under most visual encoders.

## 5 Conclusions

This paper presents VIP5, a parameter-efficient multimodal foundation recommendation model unifying vision, language, and personalization information. We design multimodal personalized prompts to integrate visual signals with text and personalization information, enhancing recommendation across diverse modalities. Our parameter-efficient tuning strategy updates a small proportion of adapters, achieving a better trade-off between recommendation performance, training efficiency, and memory usage. Through extensive experiments, we show the effectiveness of our VIP5 framework and show that multimodality information is helpful for various recommendation tasks. Future work includes further scaling up the backbone model, incorporating even more modalities, and exploring improved prompt strategies, such as chain-of-thought prompting.

## Limitations and Future Work

Despite the promising results and advantages offered by our Multimodal Foundation Model (MFM), there are several limitations that need to be addressed – 1) Bias and fairness: VIP5 relies on the quality and diversity of training data. Existing biases may lead to biased and unfair recommendations. Future work could explore methods to mitigate biases, improve data representativeness, and promote fairness of LLMs for recommendation (Li and Zhang, 2023; Hua et al., 2023a). 2) Model transparency and interpretability: VIP5 lacks inherent transparency, which can hinder users' trust in recommendations. Future work will aim to enhance transparency and explainability for VIP5-generated recommendations. 3) Scalability to other modalities: Extending the VIP5 framework to other modalities, such as audio or video, remains a challenge. Incorporating these modalities efficiently

is an important aspect for further investigation. 4) Efficiency of LLM: Efficiency is an important factor for LLMs to gain practical applications in real-world systems, because the latency should be limited to a small amount of time when delivering services to users. In this work, we have made an initial attempt to improve LLM efficiency by proposing the parameter-efficient tuning approach. In the future, it is important to investigate the efficiency of LLMs on various stages of the pipeline, such as the effiency of pre-training, fine-tuning and prompt-based inference (Li et al., 2023a). In conclusion, addressing these limitations can pave the way for improved multimodal foundation models and more effective recommendations across various applications and domains.

## Acknowledgement

This work was supported in part by NSF 1910154, 2007907, 2046457 and 2127918. Any opinions, findings, conclusions, or recommendations expressed in this material are those of the authors and do not necessarily reflect those of the sponsors.

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

# Appendix

From Figure 8 to Figure 10, we provide a detailed list of 29 multimodal personalized prompts used in our paper that covers three recommendation tasks.

---

**Prompt A-1**
**Input template:** Given the following purchase history of *user_{{user_id}}*:
*{{purchase_history}}*
predict next possible item to be purchased by the user?
**Target template:** *{{next_item}}*

**Prompt A-2**
**Input template:** I find the purchase history list of *user_{{user_id}}*:
*{{purchase_history}}*
I wonder which is the next item to recommend to the user. Can you help me decide?
**Target template:** *{{next_item}}*

---

**Prompt A-3**
**Input template:** Here is the purchase history list of *user_{{user_id}}*:
*{{purchase_history}}*
try to recommend next item to the user
**Target template:** *{{next_item}}*

**Prompt A-4**
**Input template:** Given the following purchase history of *{{user_desc}}*:
*{{purchase_history}}*
predict next possible item for the user
**Target template:** *{{next_item}}*

---

**Prompt A-5**
**Input template:** Based on the purchase history of *{{user_desc}}*:
*{{purchase_history}}*
Can you decide the next item likely to be purchased by the user?
**Target template:** *{{next_item}}*

**Prompt A-6**
**Input template:** Here is the purchase history of *{{user_desc}}*:
*{{purchase_history}}*
What to recommend next for the user?
**Target template:** *{{next_item}}*

---

**Prompt A-7**
**Input template:** *User_{{user_id}}* has the following purchase history:
*{{purchase_history}}*
Does the user likely to buy *{{item_id}}* *{{item_photo}}* next?
**Target template:** *{{answer_choices[label]}}*
*(yes/no)*

**Prompt A-8**
**Input template:** According to *{{user_desc}}*'s purchase history list:
*{{purchase_history}}*
Predict whether the user will purchase *{{item_id}}* *{{item_photo}}* next?
**Target template:** *{{answer_choices[label]}}*
*(yes/no)*

---

**Prompt A-9**
**Input template:** According to the purchase history of *{{user_desc}}*:
*{{purchase_history}}*
Can you recommend the next possible item to the user ?
**Target template:** *{{next_item}}*

Figure 8: Multimodal personalized prompts for Task Group A: Sequential Recommendation.

## Prompt B-1
**Input template:** Will *user_{{user_id}}* likely to interact with *item_{{item_id}}* *{{item_photo}}*?
**Target template:** *{{answer_choices[label]}}* *(yes/no)*

## Prompt B-2
**Input template:** Shall we recommend *item_{{item_id}}* *{{item_photo}}* to *user_{{user_id}}*?
**Target template:** *{{answer_choices[label]}}* *(yes/no)*

## Prompt B-3
**Input template:** For *{{user_desc}}*, do you think it is good to recommend *{{item_title}}* *{{item_photo}}*?
**Target template:** *{{answer_choices[label]}}* *(yes/no)*

## Prompt B-4
**Input template:** I would like to recommend some items for *user_{{user_id}}*. Is the following item a good choice? *{{item_title}}* *{{item_photo}}*
**Target template:** *{{answer_choices[label]}}* *(yes/no)*

## Prompt B-5
**Input template:** Which item of the following to recommend for *{{user_desc}}*? *{{candidate_items}}*
**Target template:** *{{target_item}}*

## Prompt B-6
**Input template:** Choose the best item from the candidates to recommend for *{{user_desc}}*? *{{candidate_items}}*
**Target template:** *{{target_item}}*

## Prompt B-7
**Input template:** Pick the most suitable item from the following list and recommend to *user_{{user_id}}*: *{{candidate_items}}*
**Target template:** *{{target_item}}*

## Prompt B-8
**Input template:** We want to make recommendation for *user_{{user_id}}*. Select the best item from these candidates: *{{candidate_items}}*
**Target template:** *{{target_item}}*

Figure 9: Multimodal personalized prompts for Task Group B: Direct Recommendation.

## Prompt C-1
**Input template:** Generate an explanation for *user_{{user_id}}* about this product: *{{item_title}}* *{{item_photo}}*
**Target template:** *{{explanation}}*

## Prompt C-2
**Input template:** Given the following review headline *{{review_headline}}* can you help generate an explanation of *user_{{user_id}}* for *item_{{item_id}}* *{{item_photo}}*?
**Target template:** *{{explanation}}*

## Prompt C-3
**Input template:** Help *user_{{user_id}}* generate a *{{star_rating}}*–star explanation about this product: *{{item_title}}* *{{item_photo}}*
**Target template:** *{{explanation}}*

## Prompt C-4
**Input template:** Generate an explanation for *{{user_desc}}* about this product: *{{item_title}}* *{{item_photo}}*
**Target template:** *{{explanation}}*

## Prompt C-5
**Input template:** Based on the following review headline: *{{review_headline}}* Generate *{{user_desc}}*'s purchase explanation about *{{item_title}}* *{{item_photo}}*
**Target template:** *{{explanation}}*

## Prompt C-6
**Input template:** Help *{{user_desc}}* generate a *{{star_rating}}*–star explanation for *item_{{item_id}}* *{{item_photo}}*
**Target template:** *{{explanation}}*

## Prompt C-7
**Input template:** Predict the star rating , then use *{{feature_word}}* as feature word to generate *user_{{user_id}}*'s purchase explanation for *item_{{item_id}}* *{{item_photo}}*
**Target template:** *{{star_rating}}*, *{{explanation}}*

## Prompt C-8
**Input template:** What score will *{{user_desc}}* rate *item_{{item_id}}* *{{item_photo}}*? Then give an explanation for the rating score. (1 being lowest and 5 being highest)
**Target template:** *{{star_rating}}*, *{{explanation}}*

## Prompt C-9
**Input template:** Based on the feature word *{{feature_word}}*, generate an explanation for *user_{{user_id}}* about this product: *{{item_title}}* *{{item_photo}}*
**Target template:** *{{explanation}}*

## Prompt C-10
**Input template:** Given the word *{{feature_word}}*, can you help generate an explanation for *{{user_desc}}* about the product: *{{item_title}}* *{{item_photo}}*
**Target template:** *{{explanation}}*

## Prompt C-11
**Input template:** Using the word *{{feature_word}}*, write a *{{star_rating}}*–star explanation for *user_{{user_id}}* about *item_{{item_id}}* *{{item_photo}}*
**Target template:** *{{explanation}}*

## Prompt C-12
**Input template:** According to the feature word *{{feature_word}}*, generate a *{{star_rating}}*–star explanation for *{{user_desc}}* about *item_{{item_id}}* *{{item_photo}}*
**Target template:** *{{explanation}}*

Figure 10: Multimodal personalized prompts for Task Group C: Explanation Generation.