# OpenReview forum: "VIP5: Towards Multimodal Foundation Models for Recommendation"
_EMNLP/2023/Conference — EMNLP 2023 Findings_

### Official Review · Reviewer_AFHM · 2023-08-03

**Paper Topic And Main Contributions:** 1. The authors introduce multimodal p…
**Soundness:** 3

**Excitement:**

3: Ambivalent: It has merits (e.g., it reports state-of-the-art results, the idea is nice), but there are key weaknesses (e.g., it describes incremental work), and it can significantly benefit from another round of revision. However, I won't object to accepting it if my co-reviewers champion it.

**Reasons To Accept:**

1. The authors introduce a unified form to model sequential recommendation, direct recommendation, and explanation using a multimodal foundation model.
2. This paper is clearly presented.
3. This paper conducts insightful ablation experiments on parameter-efficient training method and visual encoder, etc.


**Reasons To Reject:**

1. The novelty of the proposed work is somewhat limited. The parameter-efficient training method is very similar to LoRA and llama-adapter.
2. They use P5-small as the backbone model, which is a small model, and I don't understand the purpose of having to use parameter-efficient training method.
3. Experimental comparisons of fine-tuning adapters versus fine-tuning full parameters are lacking.
4. The baseline for comparison is not the latest methods.


**Reproducibility:**

3: Could reproduce the results with some difficulty. The settings of parameters are underspecified or subjectively determined; the training/evaluation data are not widely available.

**Reviewer Confidence:**

4: Quite sure. I tried to check the important points carefully. It's unlikely, though conceivable, that I missed something that should affect my ratings.

---

> ### Author Rebuttal · Authors · 2023-08-28
>
> Dear reviewer:
>
> Thanks for your comments. We address your concerns as below:
>
> Q1: The novelty of the proposed work is somewhat limited; The purpose of having to use parameter-efficient training methods; Experimental comparisons of fine-tuning adapters versus fine-tuning full parameters.
>
> A1: In this paper, we are the first to introduce parameter-efficient methods to the LLM-based generative recommendation domain. The primary consideration is that the user purchase history sequences in sequential recommendation are inherently long, and the additional visual tokens will significantly impact the model's training efficiency, even for P5-small. By incorporating parameter-efficient methods, MFM can achieve higher training efficiency while maintaining close to the original model performance, thus supporting longer prompt sequences. As shown in Figure 4, we provide a performance comparison among MFM variants and full fine-tuning. In fact, our MFM is a framework in which various parameter-efficient methods can be integrated to further improve training efficiency.
>
>
> Q2: The baseline for comparison is not the latest methods.
>
> A2: In the submission, we adopt the P5 approach that was published in September 2022 as a very recent baseline to compare.

---

### Official Review · Reviewer_tDAe · 2023-08-04

**Soundness:** 2

**Excitement:**

1: Poor: I cannot identify the contributions of this paper, or I believe the claims are not sufficiently backed up by evidence. I would fight to have it rejected.

**Missing References:**

Missing importance references:

Towards Universal Sequence Representation Learning for Recommender Systems, Hou et al, KDD'22

Selective Fairness in Recommendation via Prompts, Wu et al, SIGIR'22

Language Models as Recommender Systems: Evaluations and Limitations, Zhang et al, NeurIPS 2021 workshop on I (Still) Can’t Believe It’s Not Better

Prompt Learning for News Recommendation, Zhang et al, SIGIR'23. It was uploaded to arXiv on Apr 11 while the EMNLP deadline was on Jun 23.


**Paper Topic And Main Contributions:**

This paper adopts multimodal foundation models to exploit different modalities (text, +image+interactions) information for three recommendation-related tasks (sequential recommendation, direct recommendation, and explanation generation. It bases on the P5 model (Pretrain, Personalized Prompt & Predict Paradigm) proposed by (Geng et al, Recsys’22) and extends the P5 model to include visual modality information. Experiments on the Amazon.com real-world platform are conducted.

**Questions For The Authors:**

Q1: concerning architecture design, and experiment setup.

i) On line 362, the number of image tokens is set to 2. Is the proposed multimodal recommendation affected by this hyperparameter?

ii) On line 376, set input token’s maximum length to 1024. The size 1024 may be too limited. In SDIM (Sampling Is All You Needon Modeling Long-Term User Behaviors for CTR Prediction, Cao et al, CIKM 2022), the recent 1,024 behaviors are supported as sequence length. In such cases, the token length of 1024 is not enough. How to address the issues of capturing long-term user interests?

iii) On line 337, for the direct recommendation, we generate 100 candidate lists. Size 100 might be useful in the refining ranking stage after we have got a small set from millions of items pool. Size 100 is not directly applicable in real-world million-scale items pool. Is the proposed multimodal recommendation affected by this hyperparameter?


**Reasons To Accept:**

S1: a nice try to enable recommendation with multimodal information from the perspective of multimodal foundation models.


**Reasons To Reject:**

W1: The experimental evaluation is limited and not comprehensive.

i) The datasets are small. The proposed MFM model is evaluated on the Amazon dataset. Following the dataset link (http://jmcauley.ucsd.edu/data/amazon/links.html) given in the paper, the Amazon dataset contains 82M Ratings, 20M Users and 9M Items. However, the statistics shown in Table 1 are much smaller.

ii) the datasets are only from one recommendation platform, i.e., the Amazon.com. Actually, the recommendation datasets that contain multimodal information are abundant, including widely used benchmarks: The Yelp dataset (6M reviews + 200K pictures), and The Million Song related datasets (see Deep content-based music recommendation by van den Oord et al, NIPS’13). Furthermore, the backbone of the proposed model, i.e., the P5 model has also been evaluated on the Yelp dataset.

iii) The baselines are ignoring important SOTA methods. For sequential recommendation, IDNP: Interest Dynamics Modeling using Generative Neural Processes for Sequential Recommendation by Du et al, arXiv:2208.04600. The baselines are ignoring large proportions of works on graph neural networks (GNN)-based RecSys. See Figure 6 in [Graph Neural Networks in Recommender Systems: A Survey by Shiwen Wu et al] for more baselines. See Missing references for more works.

W2: Some claims are not sound and seem too big.

i) In the abstract, “visual information has received limited attention in recommender systems”. Typical previous works: Learning Image and User Features for Recommendation in Social Networks, ICCV'15. See the survey for more details: Recommender Systems Leveraging Multimedia Content, ACM Computing Surveys 2020.

ii) In Contribution point 1, “framework to unify CV, NLP, and RecSys foundation models”. Is the proposed framework really indeed a unified framework for CV+NLP? Or it just adopts existing NLP+CV tools for recommendation applications?

W3: concerning related works, novelty, and findings.

i) the related works section should highlight the limitations of previous (recommendation-focused) methods compared to the proposed MFM model. These differences and limitations are missing. It is needed to address the novelty issue of the proposed MFM model by thoroughly comparing it with existing methods.

ii) some findings in this paper are trivial. For example, “extensive experiments...show that multimodality information is helpful for various recommendation tasks.”

W4: The experimental results in Table 3 are unfair to the baseline P5 model. On the same three datasets (Sports, Beauty, Toys), the reported results in the P5 model’s original paper are much higher than that in this paper. Why not report the best result of P5 and explanations should be given.


**Reproducibility:**

3: Could reproduce the results with some difficulty. The settings of parameters are underspecified or subjectively determined; the training/evaluation data are not widely available.

**Reviewer Confidence:**

4: Quite sure. I tried to check the important points carefully. It's unlikely, though conceivable, that I missed something that should affect my ratings.

---

> ### Author Rebuttal · Authors · 2023-08-28
>
> Dear reviewer:
>
> Thank you for your helpful comments and suggestions. Here are our responses to your questions:
>
> Q1: The experimental evaluation is limited and not comprehensive.
>
> i) The datasets are small;  ii) the datasets are only from one recommendation platform; iii) The baselines are ignoring important SOTA methods. The baselines are ignoring large proportions of works on graph neural networks (GNN)-based RecSys.
>
> A1: For i), the 24 categories within the Amazon dataset are independent, with a combined total of 82M ratings, 20M users, and 9M items. In order to facilitate experiments and eliminate the influence of noisy users and items, we chose to use the 5-core version of the dataset for our study. This version ensures that the included users and items have k reviews each, which is a commonly adopted routine in the recommendation research community [2, 3]. Consequently, Table 1 presents the 5-core version statistics of the Amazon dataset sub-categories.
>
> For ii), though the datasets are from the Amazon platform, they are four independent datasets that represent very different item categories, i.e., sports, beauty, clothing and toys, and thus users’ behavior patterns on the four datasets can be very different. By using the four different item category datasets for experiments, we aim to validate that the proposed method is robust to different recommendation scenarios.
>
>
> For iii), we appreciate the reviewer's suggestions on incorporating additional state-of-the-art methods for comparison. In fact, our primary focus in this study is to explore the potential of language model-based generative recommendation and investigate whether multimodal information can help in improving LM-based recommendation systems.
>
> Although competing with SOTA methods is important, it is not the central focus of this paper. Nevertheless, we have compared our approach with the P5 method, which is a very recent baseline published in September 2022. In the P5 paper, the authors compared their method with SimpleX, which has been shown to outperform many GNN-based methods. Therefore, our comparison with P5 reflects our method's performance in the context of GNN-based approaches.
>
> We acknowledge that there are numerous other methods and GNN-based RecSys models that could be considered as additional baselines. In future work, we will consider incorporating more methods and further validate the effectiveness of our approach.
>
>
> Q2: Some claims are not sound and seem too big. i) In the abstract, “visual information has received limited attention in recommender systems”. ii) In Contribution point 1, “framework to unify CV, NLP, and RecSys foundation models”. Is the proposed framework really indeed a unified framework for CV+NLP? Or it just adopts existing NLP+CV tools for recommendation applications?
>
> A2: For i), our main point is that in the recommendation domain, papers utilizing image modality are relatively fewer compared to those using user behavior sequence, rating, and text review information. We acknowledge that our statement may seem subjective, and we are happy to remove this sentence in the camera-ready version. Additionally, we will cite the two papers mentioned by the reviewer.
>
> For ii), we apologize for any confusion caused by our statement. Our intention was not to claim a unified framework for CV and NLP alone, but rather a framework that combines all three domains: CV, NLP, and RecSys. Our main contribution claim is that we are the first to consider these three tasks together using a unified language modeling approach. We will revise our statement in the contribution section to better reflect this clarification.
>
>
> Q3: concerning related works, novelty, and findings. These differences and limitations are missing. It is needed to address the novelty issue of the proposed MFM model by thoroughly comparing it with existing methods.
>
> A3: Due to the 8-page limit of manuscript, we were unable to write the related work section with too much space in the paper. However, in the camera-ready version, we will have an additional page to thoroughly expand our discussion on the related work, as well as the differences and limitations of existing methods.
>
>
> Q4: The experimental results in Table 3 are unfair to the baseline P5 model. On the same three datasets (Sports, Beauty, Toys), the reported results in the P5 model’s original paper are much higher than that in this paper. Why not report the best result of P5 and explanations should be given.
>
> A4: In the original P5 paper, they used 5 task families of prompts for pretraining. However, in order to make a fair comparison with our proposed MFM model, we retrained the P5 model using prompts consistent with the 3 task groups used in our paper. This adjustment was made to ensure that both models were compared under the same conditions. We will add an explanation in the paper to clarify this point and provide a better understanding of the comparison between the MFM model and the P5 baseline.
>
> Q5: On line 362, the number of image tokens is set to 2. Is the proposed multimodal recommendation affected by this hyperparameter?
>
> A5: Indeed, the performance is affected by the number of image tokens used, which is shown in Figure 6 of our paper. In this figure, the model performance varies with different numbers of image tokens. There are trade-offs between efficiency and performance when selecting the number of image tokens. By considering these trade-offs, we have chosen 2 image tokens for our experiments, which provides a good balance between computational efficiency and recommendation performance.
>
>
> Q6: The size 1024 may be too limited. In such cases, the token length of 1024 is not enough. How to address the issues of capturing long-term user interests?
>
> A6: It is important to note that this limitation is not specific to our proposed method but a common issue for any large language model (LLM) with a fixed context length. In our experiments, we were constrained by the available GPU resources, and we found that the input length of 1024 was sufficient for the dataset we adopted. However, it is possible to increase the context length if more powerful GPUs are used, or if other techniques for handling longer sequences are employed. Furthermore, our proposed parameter-efficient method has already helped to increase the token length: by incorporating parameter-efficient methods, MFM can achieve higher training efficiency while maintaining close to the original model performance, thus supporting longer prompt sequences.
>
>
> Q7: Size 100 is not directly applicable in a real-world million-scale items pool. Is the proposed multimodal recommendation affected by this hyperparameter?
>
> A7: The size 100 is not a hyperparameter in our model but rather a widely used experimental setup in many existing research works (e.g.,  CF [1], SASRec [2], S3-Rec [3], BERT4Rec [4]). In a real-world large-scale scenario, our proposed multimodal recommendation approach can be used effectively in the refining stage. For instance, initial candidate items can first be generated using simple recommendation techniques, reducing the million-scale item pool to a smaller, more manageable size. Our approach can then be applied to further refine and rank these items for the final recommendation list, taking advantage of the multimodal information.
>
>
> Q8: Missing important references.
>
> A8: Thanks for pointing out these references. We recognize the importance of including relevant and recent works in our paper to provide a comprehensive understanding of the research landscape. We will include the references in the camera-ready version of our paper.
>
>
> Reference:
>
> [1] Koren, Yehuda. "Factorization meets the neighborhood: a multifaceted collaborative filtering model." Proceedings of the 14th ACM SIGKDD international conference on Knowledge discovery and data mining. 2008.
>
> [2] Kang, Wang-Cheng, and Julian McAuley. "Self-attentive sequential recommendation." 2018 IEEE international conference on data mining (ICDM). IEEE, 2018.
>
> [3] Zhou, Kun, et al. "S3-rec: Self-supervised learning for sequential recommendation with mutual information maximization." Proceedings of the 29th ACM international conference on information & knowledge management. 2020.
>
> [4] Sun, Fei, et al. "BERT4Rec: Sequential recommendation with bidirectional encoder representations from transformer." Proceedings of the 28th ACM international conference on information and knowledge management. 2019.

---

### Official Review · Reviewer_hNgC · 2023-08-09

**Soundness:** 3

**Excitement:**

3: Ambivalent: It has merits (e.g., it reports state-of-the-art results, the idea is nice), but there are key weaknesses (e.g., it describes incremental work), and it can significantly benefit from another round of revision. However, I won't object to accepting it if my co-reviewers champion it.

**Paper Topic And Main Contributions:**

The key contributions are
A) The multimodal foundation model framework
B) A parameter-efficient training method
C) Experimental results

**Reasons To Accept:**

The framework performs well in three types of recommendation tasks: sequential, direct and explanation generation

**Reasons To Reject:**

I don't see any weakness

**Reproducibility:**

3: Could reproduce the results with some difficulty. The settings of parameters are underspecified or subjectively determined; the training/evaluation data are not widely available.

**Reviewer Confidence:**

3: Pretty sure, but there's a chance I missed something. Although I have a good feel for this area in general, I did not carefully check the paper's details, e.g., the math, experimental design, or novelty.

---

> ### Author Rebuttal · Authors · 2023-08-28
>
> Dear reviewer:
>
> Thank you for the positive comments!

---

### Official Review · Reviewer_gRHd · 2023-08-12

**Typos Grammar Style And Presentation Improvements:** na
**Soundness:** 3

**Excitement:**

4: Strong: This paper deepens the understanding of some phenomenon or lowers the barriers to an existing research direction.

**Missing References:**

na

**Paper Topic And Main Contributions:**

This paper introduces the Multimodal Foundation Model (MFM) framework, which serves to integrate foundational models from Computer Vision (CV), Natural Language Processing (NLP), and Recommender Systems (RecSys). The framework aims to enhance recommendations by leveraging multimodal information. The method, in particular, attains remarkable performance and operational efficiency across three distinct task groups. This achievement is attributed to the integration of features like multimodal personalized prompts and a parameter-efficient training approach.

**Questions For The Authors:**

na

**Reasons To Accept:**

1.The related work is well-presented, and the illustration of the MFM framework is clear and concise.
2.The comprehensive performance comparison showcased in this paper highlights a significant enhancement, thereby offering robust substantiation of the effectiveness of the proposed algorithm.


**Reasons To Reject:**

1.Could the authors provide further clarification regarding the distinctions among prompts such as A-1 to A-3 in the Appendix? Are these differences primarily confined to minor variations in wording, and if so, what is the significance of adopting such an approach?
2.Figure 6 presents challenges in terms of discerning patterns, particularly the notably inferior performance of the 3-image tokens. The paper would benefit from an explanation of this unexpected observation.
3.The contrast in performance between different visual encoders, specifically B-8 and C-12 in Figure 7, appears striking. Relating this solely to compatibility issues, as mentioned in the text, might not offer a wholly convincing explanation.

**Reproducibility:**

3: Could reproduce the results with some difficulty. The settings of parameters are underspecified or subjectively determined; the training/evaluation data are not widely available.

**Reviewer Confidence:**

4: Quite sure. I tried to check the important points carefully. It's unlikely, though conceivable, that I missed something that should affect my ratings.

---

> ### Author Rebuttal · Authors · 2023-08-28
>
> Dear reviewer:
>
> Thanks for your comments and questions. We address your concerns as below:
>
> Q1: Are these differences primarily confined to minor variations in wording, and if so, what is the significance of adopting such an approach?
>
> A1: Yes. We design these multimodal personalized prompts to express similar sequential recommendation instructions using different wording. This prevents the model from overfitting to a specific instruction during training. In fact, the diversity of prompts is crucial -- having more variations of prompts allows the model’s attention to focus more on the useful information in the input, such as the user's purchase history, to generate the desired output, rather than other ordinary words in the prompts which serve as instructions. When we developed our approach, we also found that if we reduced the number of multimodal personalized prompts during training, the performance of the model would decrease as well.
>
>
> Q2: Explanation of the notably inferior performance of the 3 image tokens in Figure 6.
>
> A2: The difference in values for the 3-image tokens scenario is not as large as shown in the graph. Since the values are relatively close to each other, the gap between the Y-axis markings for NDCG@5 is very small. In Figure 6, the four bar values in the B-8 scenario are 0.0661 (1 image token), 0.0671 (2 image tokens), 0.0649 (3 image tokens),  0.0687 (5 image tokens).
>
>
> Q3:  Explanation of the contrast in performance between different visual encoders, specifically B-8 and C-12 in Figure 7.
>
> A3: As we mentioned in the paper, visual information has a positive correlation with user choices in direct recommendation situations, which is why the trend observed in Task B-8 in Figure 7 is in line with our expectations. However, when it comes to hint word based explanation generation tasks, not all aspects that users focus on are related to appearance. For instance, sometimes users need explanations about whether a tool is durable or if a game is enjoyable. It's important to note that this task is different from a pure image captioning task, as the focus is not solely on describing the visual aspects of an item. Therefore, it is quite normal for the trend not to follow the order of better visual encoders if the ground-truth explanation does not have a direct correlation with visual information.

---

### Meta-Review · Senior_Area_Chairs · 2023-10-05

**Recommendation:** 3

**Metareview:**

The reviewers appreciated the incorporation of multimodal information for recommendation and the efficiency of the method in terms of performance. However, the authors, are highly encouraged to include their clarifications during the rebuttal in the final version of the paper and:

- clarify the writing as suggested by R3: avoid any potential overclaim, revise the related work section and situate this work a bit better,

- add experiments (e.g., those suggested by R3 and R4),

- better explain the motivation for parameter efficiency.

---

### Decision · Program_Chairs · 2023-10-07

**Decision:**

Accept-Findings

**Comment:**

The reviewers appreciated the incorporation of multimodal information for recommendation and the efficiency of the method in terms of performance. However, the authors, are highly encouraged to include their clarifications during the rebuttal in the final version of the paper and:

- clarify the writing as suggested by R3: avoid any potential overclaim, revise the related work section and situate this work a bit better,

- add experiments (e.g., those suggested by R3 and R4),

- better explain the motivation for parameter efficiency.